# Non-Uniform Smoothness for Gradient Descent

**Albert S. Berahas**                                      *albertberahas@gmail.com*
*Department of Industrial and Operations Engineering*
*University of Michigan*

**Lindon Roberts**                                     *lindon.roberts@sydney.edu.au*
*School of Mathematics and Statistics*
*University of Sydney*

**Fred Roosta**                                            *fred.roosta@uq.edu.au*
*School of Mathematics and Physics*
*University of Queensland*
*ARC Training Centre for Information Resilience*

**Reviewed on OpenReview:** *https: // openreview. net/ forum? id= 17ESEjETbP*

## Abstract

The analysis of gradient descent-type methods typically relies on the Lipschitz continuity of
the objective gradient. This generally requires an expensive hyperparameter tuning process
to appropriately calibrate a stepsize for a given problem. In this work we introduce a local
first-order smoothness oracle (LFSO) which generalizes the Lipschitz continuous gradients
smoothness condition and is applicable to any twice-differentiable function. We show that
this oracle can encode all relevant problem information for tuning stepsizes for a suitably
modified gradient descent method and give global and local convergence results. We also
show that LFSOs in this modified first-order method can yield global linear convergence
rates for non-strongly convex problems with extremely flat minima.

## 1 Introduction

In this work, we consider gradient descent-type algorithms for solving unconstrained optimization problems

$$\min_{x \in \mathbb{R}^d} f(x), \tag{1.1}$$

where $f : \mathbb{R}^d \to \mathbb{R}$ is continuously differentiable. Gradient descent-type methods for (1.1) have seen renewed
interest arising from applications in data science, where $d$ is potentially very large (Bottou et al., 2018;
Wright, 2018). When analyzing such methods with a fixed stepsize, it is most commonly assumed that $\nabla f$ is
$L_f$-Lipschitz continuous and assume a stepsize of order $1/L_f$, e.g. (Wright, 2018, Section 4). The key reason
behind this smoothness assumption is the resulting bound

$$|f(y) - f(x) - \nabla f(x)^T(y - x)| \leq \frac{L_f}{2}\|y - x\|_2^2, \qquad \forall x, y \in \mathbb{R}^d.$$

In practice, however, $L_f$ is usually not known. The simplest approach in this situation is to try several
stepsize choices as part of a (possibly automated) hyperparameter tuning process (Snoek et al., 2012). That
said there are often benefits to changing the stepsize at each iteration, with common approaches including
linesearch methods (Nocedal & Wright, 2006), Polyak's stepsize for convex functions (Polyak, 1969) (although
this requires knowledge of the global minimum of $f$) and the Barzilai-Borwein stepsize (Barzilai & Borwein,
1988) (although global convergence theory exists only for strictly convex quadratic functions, and does not
converge for some (non-quadratic) strongly convex functions without modification (Burdakov et al., 2019)).

In the setting where $L_f$ is known, this approach is still limited pessimistic, as the stepsize is chosen to be sufficiently small to ensure decrease throughout the domain, as it reflects the global smoothness of $f$. In an iterative method it suffices to ensure decrease from one iterate to the next, and so of greater relevance is the *local smoothness* of $f$ in a neighborhood of the current iterate.

There are two natural ways that local smoothness could be used to modify the stepsizes in gradient descent. First, we could dynamically estimate $L_f$ by looking at how rapidly $\nabla f$ is changing near the current iterate. This approach is followed in (Malitsky & Mishchenko, 2020), where $L_f$ is estimated based on $\|\nabla f(x_k) - \nabla f(x_{k-1})\|_2 / \|x_k - x_{k-1}\|_2$. This estimation procedure allows for global convergence of a hyperparameter-free variant of gradient descent for convex functions. An alternative approach for convex problems is to estimate the distance from the starting point to the minimizer $\|x_0 - x^*\|$ and use this to construct hyperparameter-free methods (Carmon & Hinder, 2022; Defazio & Mishchenko, 2023; Mishchenko & Defazio, 2023). Alternatively, we could assume the existence of an oracle that gives us a suitable local Lipschitz constant for $\nabla f$ (Mei et al., 2021). There, it is assumed that one has access to an oracle $L(x)$ that satisfies

$$|f(y) - f(x) - \nabla f(x)^T(y-x)| \leq \frac{L(x)}{2}\|y-x\|_2^2, \qquad \forall y \in \mathbb{R}^d. \tag{1.2}$$

Although some useful properties arise when allowing a stepsize of $1/L(x_k)$ for gradient descent (at current iterate $x_k$)—for example faster convergence than gradient descent with fixed stepsizes—the oracle $L(x)$ has limitations. For example, if we want to exploit extremely flat regions around minima, we may want $L(x) \to 0$ as $x \to x^*$, but for (1.2) to hold with $L(x^*) = 0$ requires $f$ be a constant function.

In a separate line of work, (Zhang et al., 2020) considers gradient clipping methods applied to functions satisfying a smoothness condition of the form $\|\nabla^2 f(x)\|_2 \leq L_0 + L_1\|\nabla f(x)\|_2$, which was empirically motivated by training deep neural networks and also applies to some distributionally robust optimization problems (Jin et al., 2021). This condition was slightly relaxed in (Xie et al., 2023) to the generalized Lipschitz condition

$$\|\nabla f(y) - \nabla f(x)\|_2 \leq (L_0 + L_1\|\nabla f(x)\|_2) \cdot \|y-x\|_2, \quad \forall x \in \mathbb{R}^d, \ \forall y \in B(x, 1/L_1), \tag{1.3}$$

noting in particular that the inequality only needs to hold for $x$ and $y$ sufficiently close.

In this work, we consider a more general and localized form of the approach from (Mei et al., 2021), where our oracle only requires bounds similar to (1.2) to hold for $y \in B(x, R)$ for some $R > 0$. Such a local first-order smoothness oracle (LFSO), gives a local Lipschitz constant over an explicitly defined neighborhood. Such LFSOs exist for any $C^2$ function (not just those with Lipschitz continuous gradients), including those satisfying the weaker smoothness condition from (Zhang et al., 2020; Jin et al., 2021) mentioned above. Given the availability of a LFSO, we show how this can be used to construct a convergent gradient descent-type iteration that does not require any hyperparameter tuning. Similar to the case of $1/L_f$ stepsizes, this parameter-free result comes from the LFSO capturing all the relevant problem information.

To show the promise of this approach, we demonstrate in Section 4 how the incorporation of a LFSO allows gradient descent to achieve a *global* linear convergence rate to arbitrarily flat minima of convex functions. We do this by considering a class of objective functions which include $f(x) = \|x\|_2^{2p}$ for $p \to \infty$, where $f(x)$ is extremely flat near the minimizer $x^* = 0$. This is achieved by the LFSO allowing larger stepsizes as $x \to x^*$, counteracting the degeneracy of the minimizer. These functions are known to be difficult for gradient descent iterations, and in fact provide the worst-case lower bounds for accelerated gradient descent (Attouch et al., 2018; 2022).[1] Further, in Section 5 we show that our LFSO approach gives global linear convergence rates for objectives of the form $f(x) = \|x\|_{2p}^{2p}$ for $p \to \infty$, an alternative set of objectives with very degenerate minimizers. Our numerical results in Section 6 show the global linear rate achieved in practice by the LFSO approach for both types of objective functions, compared to the (expected) sublinear rate for standard gradient descent. We note that the objective $f(x) = \|x\|_{2p}^{2p}$ can alternatively be minimized in 1 iteration using iteratively reweighted least-squares (IRLS). IRLS only has local convergence guarantees for problems of the form $f(x) = \|Ax - b\|_{2p}^{2p}$ when $p < 3/2$ and can diverge for larger $p$ without careful modification (Adil et al., 2019).

---

[1]Of course, LFSO requires more problem information than these methods to achieve this result.

Compared to the similar methods mentioned above, LFSO is the only method with global and local convergence theory for general $C^2$ nonconvex objectives. By comparison, (Malitsky & Mishchenko, 2020; Carmon & Hinder, 2022; Defazio & Mishchenko, 2023; Mishchenko & Defazio, 2023) only have global convergence results for convex objectives[2], and (Mei et al., 2021) only considers global convergence for nonconvex functions in the presence of a specific non-uniform Łojasiewicz condition. Our numerical results for specific convex functions in Section 6 show that the method from (Malitsky & Mishchenko, 2020) gives fast global linear convergence on the same problems, but this is not proven. We also note that the LFSO approach is potentially more suited for adaptation to stochastic optimization than (Malitsky & Mishchenko, 2020) as it does not require taking differences of (stochastic) gradients.

In the general nonconvex case with Lipschitz continuous gradients, LFSO recovers the standard $\mathcal{O}(\epsilon^{-2})$ iteration complexity required to find an $\epsilon$-optimal point. The per-iteration cost of LFSO is one gradient evaluation and up to two LFSO evaluations. In terms of scalability, the practicality of this approach depends on how efficient an LFSO evaluation is; identifying general settings where LFSOs are easily computable is an important direction for future work. However, for all the example problems considered in this work, the computational cost of evaluating the LFSO is similar to (or cheaper than) the cost of a single gradient evaluation. Our approach also avoids the need for any expensive tuning of learning rates (and in some cases achieving global linear convergence rates).

This introduction and first analysis of LFSO methods demonstrates it to be a promising avenue for future work, by giving an explicit mechanism to decoupling stepsize selection from objective smoothness (and hence avoiding hyperparameter tuning), and in some specific circumstances being able to adapt to extremely flat regions of the objective function. The LFSO approach requires a more complex oracle and a choice of initial radii $R_k$ (for examples in Sections 4 and 5, $R_k = \|\nabla f(x_k)\|_2$ is used), and so an important starting point for future work would be the efficient construction of such oracles (and indeed it would be useful to know when such oracles are easy to compute), and developing a systematic understanding of suitable values for $R_k$. This future work would enable a thorough numerical evaluation of the LFSO approach on more realistic problems.

## 2 Algorithmic Framework

We now introduce our new oracle and show our algorithmic framework which incorporates the LFSO oracle into a gradient descent-type method.

**Definition 2.1.** *Suppose $f : \mathbb{R}^d \to \mathbb{R}$ is continuously differentiable. A function $L : \mathbb{R}^d \times (0, \infty) \to [0, \infty)$ is a* local first-order smoothness oracle (LFSO) *for $f$ if*

$$|f(y) - f(x) - \nabla f(x)^T(y - x)| \leq \frac{L(x, R)}{2}\|y - x\|^2, \tag{2.1}$$

*for all $x \in \mathbb{R}^d$, all $y \in B(x, R)$ and for some $R > 0$, and $L$ is non-decreasing in the second argument $R$.*

For example, if $f$ is $L_f$-smooth—i.e., $\nabla f$ is Lipschitz continuous with constant $L_f$—then $L(x, R) = L_f$ defines an LFSO for $f$. Treating this as an oracle rather than a problem constant encodes the common algorithmic assumption that $L_f$ is known. However, LFSOs exist for a much broader class than $L_f$-smooth functions.

**Lemma 2.2.** *If $f$ is $C^2(\mathbb{R}^d)$, then $L(x, R) = \max_{y \in B(x,R)} \|\nabla^2 f(y)\|$ is an LFSO for $f$.*

*Proof.* It is straightforward that $L$ is non-decreasing in $R$. In this case, the property (2.1) is well-known, for instance (Cartis et al., 2022, Corollary A.8.4 & Theorem A.8.5) □

Another common smoothness class is the case of $C^2$ functions with Lipschitz continuous Hessians. In this case, Lemma 2.2 gives us a simple, explicit form for an LFSO.

---

[2]Furthermore, (Carmon & Hinder, 2022) requires uniformly bounded (stochastic) (sub)gradients, and (Defazio & Mishchenko, 2023; Mishchenko & Defazio, 2023) require $f$ to be Lipschitz continuous.

**Lemma 2.3.** *If $f$ is $C^2(\mathbb{R}^d)$ and $\nabla^2 f$ is $L_H$-Lipschitz continuous, then $L(x, R) = \|\nabla^2 f(x)\| + L_H R$ is an LFSO for $f$.*

*Proof.* This follows from Lemma 2.2 and that $\|\nabla^2 f(y)\| \leq \|\nabla^2 f(x)\| + L_H \|x - y\|$ for all $y \in B(x, R)$. $\quad\square$

A useful property that we will use to get explicit formulae for LFSOs is the following straightforward consequence of (2.1).

**Lemma 2.4.** *Suppose $L$ is a LFSO for $f$ and $\tilde{L}(x, R) \geq L(x, R)$ for all $x \in \mathbb{R}^d$ and all $R > 0$. Then $\tilde{L}$ is an LFSO for $f$ if and only if $\tilde{L}$ is non-decreasing in $R$.*

To incorporate an LFSO into a gradient descent-like iteration, a natural step would be to consider an iteration of the form

$$x_{k+1} = x_k - \frac{\eta}{L(x_k, R_k)} \nabla f(x_k), \tag{2.2}$$

for some appropriately chosen values of $R_k > 0$ and $\eta > 0$. In the case where $f$ is $L_f$-smooth, standard convergence theory (e.g., (Wright, 2018, Theorem 4.2.1)) would suggest that a value such as $\eta = 1$ is appropriate. This shows that the LFSO notion encapsulates the problem-specific aspect of choosing a suitable stepsize.

However, the choice of $R_k$ is not so straightforward: for property (2.1) to be useful—typically to prove that $f(x_{k+1}) < f(x_k)$—we would need $x_{k+1} \in B(x_k, R_k)$. This is not guaranteed if $R_k$ is chosen too small, as the following example illustrates.

*Example* 2.5. If $d = 1$ and $f(x) = x^4$ then a suitable LFSO is

$$L(x, R) = 24x^2 + 24R^2,$$

using Lemmas 2.2 and 2.4, as well as

$$\max_{|y - x| \leq R} 12y^2 = 12 \max\{(x - R)^2, (x + R)^2\} \leq 12(2x^2 + 2R^2),$$

from Young's inequality. Suppose $x_k = 1$ and $\eta = 1$, then from (2.2) we may compute

$$|x_{k+1} - x_k| = \frac{1}{6 + 6R_k^2}.$$

and so $x_{k+1} \notin B(x_k, R_k)$ whenever $R_k$ is smaller than $R^* \approx 0.16238$, the unique real root of $p(R) = 6R^3 + 6R^2 - 1$.

To avoid this issue, we first pick an arbitrary value for $R_k$, then possibly increase it to a sufficiently large value that $x_{k+1}$ is in the required neighborhood of $x_k$. The resulting method is given in Algorithm 1. Note that it requires one evaluation of $\nabla f$ and possibly two evaluations of $L$ (oracle calls) per iteration.

**Computational Cost of LFSO Evaluation** In practice, the computational cost of $L$ (performed potentially twice per iteration of Algorithm 1) will depend on the specific problem being analyzed. However, for the examples in this work the cost of one LFSO evaluation is small, of the same order as a single gradient evaluation (or potentially cheaper if the gradient has already been evaluated).

In Section 4, where $f(x) = h(g(x))$, the dominant cost of the LFSO (4.2) is the evaluation of $\|\nabla g(x)\|$, but $\nabla g(x)$ has already been evaluated when originally computing $\nabla f(x)$.

In Section 5 for linear regression problems, provided quantities only depending on the problem parameters $A$ have been pre-computed, the LFSO (5.3) is dominated by one residual calculation $Ax - b$. This residual is computed when evaluating the gradient, so the LFSO cost is dominated by evaluating $\|Ax - b\|_\infty$ when $Ax - b$ is already known, of cost $\mathcal{O}(n)$ for a problem with $n$ residuals. This is cheaper than the cost of a gradient evaluation, which also requires a matrix-vector product not present in the LFSO evaluation.

---

**Algorithm 1** Gradient Descent with LFSO.

---

**Input:** Starting point $x_0 \in \mathbb{R}^d$, stepsize factor $\eta > 0$.

1: **for** $k = 0, 1, 2, \ldots$ **do**
2:     Choose any $R_k > 0$
3:     Set

$$\widetilde{R}_k := \max \left\{ R_k, \frac{\eta}{L(x_k, R_k)} \|\nabla f(x_k)\| \right\}. \tag{2.3}$$

4:     Iterate

$$x_{k+1} = x_k - \frac{\eta}{L(x_k, \widetilde{R}_k)} \nabla f(x_k). \tag{2.4}$$

5: **end for**

---

In Appendix A, we also derive an LFSO for 2-class logistic regression and show the same cost as Section 5: the dominant cost of evaluating the LFSO is an extra $\mathcal{O}(n)$ for training data of length $n$ (beyond information already computed in a gradient evaluation), plus gradient evaluation requires an extra matrix-vector product not present in the LFSO evaluation.

## 3   Convergence Analysis

In this section we provide global convergence results for the general nonconvex and PL/strongly convex cases, and also local convergence rates to non-degenerate local minima. To enable these, we prove a descent lemma (Lemma 3.2) suitable for Algorithm 1.

**Assumption 3.1.** *The function $f : \mathbb{R}^d \to \mathbb{R}$ is continuously differentiable and bounded below by $f^*$, and $L$ is an LFSO for $f$.*

**Lemma 3.2.** *If Assumption 3.1 holds, then Algorithm 1 produces iterates satisfying*

$$f(x_{k+1}) \leq f(x_k) - \frac{\eta}{L(x_k, \widetilde{R}_k)} \left(1 - \frac{\eta}{2}\right) \|\nabla f(x_k)\|^2, \tag{3.1}$$

*for all $k = 0, 1, \ldots$.*

*Proof.* Since $\widetilde{R}_k \geq R_k$ and $L$ is non-decreasing in $R$, we have that

$$\|x_{k+1} - x_k\| = \frac{\eta}{L(x_k, \widetilde{R}_k)} \|\nabla f(x_k)\| \leq \frac{\eta}{L(x_k, R_k)} \|\nabla f(x_k)\| \leq \widetilde{R}_k, \tag{3.2}$$

and so (2.1) can be used. That is,

$$f(x_{k+1}) - f(x_k) \leq -\frac{\eta}{L(x_k, \widetilde{R}_k)} \|\nabla f(x_k)\|^2 + \frac{L(x_k, \widetilde{R}_k)}{2} \left( \frac{\eta^2}{L(x_k, \widetilde{R}_k)^2} \|\nabla f(x_k)\|^2 \right),$$
$$= -\frac{\eta}{L(x_k, \widetilde{R}_k)} \left(1 - \frac{\eta}{2}\right) \|\nabla f(x_k)\|^2,$$

which gives the desired result. □

Since the LFSO captures all the necessary problem information, the requirements on the stepsize factor $\eta$ are straightforward.

### 3.1 Global Convergence

**Theorem 3.3.** *If Assumption 3.1 holds and $0 < \eta < 2$, then either*

$$\liminf_{k \to \infty} \|\nabla f(x_k)\| = 0, \qquad or \qquad \lim_{k \to \infty} L(x_k, \widetilde{R}_k) = \infty.$$

*Proof.* From Lemma 3.2, we have

$$f(x_k) - f(x_{k+1}) \geq \frac{\eta(2-\eta)}{2L(x_k, \widetilde{R}_k)} \|\nabla f(x_k)\|^2. \tag{3.3}$$

By summing over $k$ we get

$$\sum_{k=0}^{\infty} \frac{\|\nabla f(x_k)\|^2}{L(x_k, \widetilde{R}_k)} \leq \frac{2[f(x_0) - f^*]}{\eta(2-\eta)} < \infty, \tag{3.4}$$

hence $\lim_{k \to \infty} \|\nabla f(x_k)\|^2 / L(x_k, \widetilde{R}_k) = 0$, and the result follows. □

Of course, this is not quite a convergence proof, as we need to be concerned about the risk of $L(x_k, \widetilde{R}_k)$ growing unboundedly, which could occur in cases such as $f(x) = \sin(x^2)$ if $\|x_k\| \to \infty$ during the iteration. One simple situation where this behavior does not occur is the following.

**Corollary 3.4.** *Suppose Assumption 3.1 holds and $0 < \eta < 2$. If the sublevel set $\{x \in \mathbb{R}^d : f(x) \leq f(x_0)\}$ is bounded, $L$ is continuous in $x$, and the choices $R_k$ are bounded, then $\lim_{k \to \infty} \|\nabla f(x_k)\| = 0$.*

*Proof.* If $R_k \leq R$ for all $k$, then $L(x_k, R_k) \leq L(x_k, R)$. From Lemma 3.2, we know $x_k \in \{x \in \mathbb{R}^d : f(x) \leq f(x_0)\}$ for all $k$. Then we know that $R_k$, $\|\nabla f(x_k)\|$ and $L(x_k, R_k)$ are all bounded, and so $\widetilde{R}_k$ is bounded too. Finally, this means $L(x_k, \widetilde{R}_k)$ is bounded, so $\lim_{k \to \infty} \|\nabla f(x_k)\|^2 / L(x_k, \widetilde{R}_k) = 0$ (derived in the proof of Theorem 3.3) gives the result. □

In fact, under the assumptions of Corollary 3.4, (3.4) actually gives us the common $\mathcal{O}(\epsilon^{-2})$ worst-case iteration complexity rate[3] for nonconvex problems (e.g., (Cartis et al., 2022, Chapter 2)). We note that these assumptions are weaker than assuming $L_f$-smoothness everywhere, as we only care about the LFSO in the initial sublevel set.

*Remark* 3.5. The above (in particular (3.2)) still works if we replace $\|\nabla f(x_k)\|$ in (2.3) with any upper bound $C_k \geq \|\nabla f(x_k)\|$. This will be useful in Section 5.

In the case where $f$ satisfies the Polyak-Łojasiewicz (PL) inequality with parameter $\mu > 0$—for example if $f$ is $\mu$-strongly convex—we can achieve convergence even in some cases where $L(x_k, \widetilde{R}_k) \to \infty$, provided it does not increase too quickly.

**Theorem 3.6.** *Suppose Assumption 3.1 holds and $0 < \eta < 2$. If $f$ is $\mu$-PL, that is*

$$f(x) - f^* \leq \frac{1}{2\mu} \|\nabla f(x)\|^2, \qquad \forall x \in \mathbb{R}^d, \tag{3.5}$$

*and*

$$\sum_{k=0}^{\infty} \frac{1}{L(x_k, \widetilde{R}_k)} = \infty,$$

*then $\lim_{k \to \infty} f(x_k) = f^*$.*

---

[3]i.e., the maximum number of iterations before $\|\nabla f(x_k)\| \leq \epsilon$ is first attained.

*Proof.* From Lemma 3.2 and (3.5), we get

$$f(x_{k+1}) - f^* \leq f(x_k) - f^* - \frac{\eta}{L(x_k, \widetilde{R}_k)}\left(1 - \frac{\eta}{2}\right)\|\nabla f(x_k)\|^2, \tag{3.6}$$

$$\leq \left[1 - \frac{\mu\eta(2 - \eta)}{L(x_k, \widetilde{R}_k)}\right](f(x_k) - f^*). \tag{3.7}$$

By summing over $k$ we get, by the assumption in the theorem statement,

$$\sum_{k=0}^{\infty} \frac{\mu\eta(2 - \eta)}{L(x_k, \widetilde{R}_k)} = \infty, \tag{3.8}$$

and so the result follows from (Bertsekas, 2015, Prop A.4.3). □

We should note that if $f$ is coercive (e.g. if $f$ is strongly convex) then the sublevel set $\{x : f(x) \leq f(x_0)\}$ is bounded, and so Corollary 3.4 guarantees convergence (provided the $R_k$ are bounded).

## 3.2 Local Convergence Rate

Encouraged by the result in Theorem 3.6, we now consider the local convergence rate of Algorithm 1 to non-degenerate local minimizers.

**Theorem 3.7.** *Suppose Assumption 3.1 holds and $0 < \eta < 2$. Suppose also that $f$ is also $C^2(\mathbb{R}^d)$ and $x^*$ is a local minimizer of $f$ with $\lambda_{\min}(\nabla^2 f(x^*)) > 0$, $L$ is continuous in $x$, and $R_k > 0$ for all $k$. If $x_0$ is sufficiently close to $x^*$ in the sense that*

- *$\|x_0 - x^*\| \leq R_1$ (where $B(x^*, R_1)$ is a region where $f$ is $\mu := \lambda_{\min}(\nabla^2 f(x^*))/2$-strongly convex), and*

- *$f(x_0) - f(x^*) \leq \mu R_2^2/2$ (where $R_2 \leq R_1/2$ and $\|\nabla f(x)\| \leq \mu R_1/(2\eta)$ whenever $x \in B(x^*, R_2)$)*

*then $x_k \to x^*$ R-linearly, with*

$$\|x_k - x^*\|^2 \leq \frac{L_{\max}}{\mu}\left(1 - \frac{\mu\eta(2 - \eta)}{L_{\max}}\right)^k \|x_0 - x^*\|^2, \tag{3.9}$$

*for constant $L_{\max} > 0$ (defined in (3.10) below).*

*Proof.* Since $f$ is $C^2$, there exists a neighborhood $B(x^*, R_1)$ within which $f$ is $\mu$-strongly convex for $\mu := \lambda_{\min}(\nabla^2 f(x^*))/2$. Given this neighborhood, define

$$L_{\max} := \max_{x \in B(x^*, R_1)} \max_{0 \leq R \leq R_1} L(x, R). \tag{3.10}$$

Hence, whenever $\|x_k - x^*\| \leq R_1$, we have $L(x_k, R_k) \leq L_{\max}$. Strong convexity also gives

$$f(y) - f(x) - \nabla f(x)^T(y - x) \geq \frac{\mu}{2}\|y - x\|^2,$$

for any $x, y \in B(x^*, R_1)$, and so it follows that $L(x, R) \geq \mu$ for all $x \in B(x^*, R_1)$ and $R > 0$.

Since $f$ is $C^1$, there exists[4] an $R_2 \leq R_1/2$ such that $\|\nabla f(x)\| \leq \mu R_1/(2\eta)$ for all $x \in B(x^*, R_2)$. Then if $\|x_k - x^*\| \leq R_2$, we have

$$\|x_{k+1} - x^*\| \leq \|x_k - x^*\| + \frac{\eta}{L(x_k, \widetilde{R}_k)}\|\nabla f(x_k)\| \leq R_2 + \frac{R_1}{2} \leq R_1.$$

---

[4]Since $\nabla f$ is continuous, there is a ball $B(x^*, R_2')$ such that $\|\nabla f(x)\| \leq \epsilon$ for all $x \in B(x^*, R_2')$ (with $\epsilon$ arbitrary, such as $\mu R_1/(2\eta)$ as above). Then take $R_2 = \min(R_1/2, R_2')$.

Now, for any $x \in B(x^*, R_1)$, by strong convexity in this region, it follows that if $f(x) - f(x^*) \leq \mu R_2^2/2$ then $\|x - x^*\| \leq R_2$. Given this, suppose that $x_0$ is sufficiently close to $x^*$ in the sense that $x_0 \in B(x^*, R_1)$ and $f(x_0) - f(x^*) \leq \mu R_2^2/2$. We then have that $x_0 \in B(x^*, R_2)$ and so $x_1 \in B(x^*, R_1)$ from the above. Lemma 3.2 implies that $f(x_1) \leq f(x_0)$ and so $x_1$ also satisfies $f(x_1) - f(x^*) \leq \mu R_2^2/2$, which in turn implies $x_1 \in B(x^*, R_2)$.

By induction, we conclude that $x_k \in B(x^*, R_1)$ for all $k$ (and also that $x_k \in B(x^*, R_2)$ for all $k$). Then, by the same reasoning as (3.7), since (3.5) holds in $B(x^*, R_1)$ and $L(x_k, \widetilde{R}_k) \leq L_{\max}$, we have

$$f(x_{k+1}) - f(x^*) \leq \left(1 - \frac{\mu\eta(2-\eta)}{L_{\max}}\right)(f(x_k) - f(x^*)),$$

and we have a linear convergence rate of $f(x_k) \to f(x^*)$.

Finally, we use the strong convexity result that (Nesterov, 2004, Lemma 1.2.3 & Theorem 2.1.7)

$$\frac{\mu}{2}\|x - x^*\|^2 \leq f(x) - f(x^*) \leq \frac{L_{\max}}{2}\|x - x^*\|^2,$$

to conclude that

$$\|x_k - x^*\|^2 \leq \frac{2}{\mu}(f(x_k) - f(x^*)) \leq \frac{2}{\mu}\left(1 - \frac{\mu\eta(2-\eta)}{L_{\max}}\right)^k (f(x_0) - f(x^*))$$

$$\leq \frac{L_{\max}}{\mu}\left(1 - \frac{\mu\eta(2-\eta)}{L_{\max}}\right)^k \|x_0 - x^*\|^2,$$

giving (3.9). $\qquad\square$

## 4 Global Rates for Compositions of Functions

Motivated by problems with very flat minima, we now consider the performance of Algorithm 1 when the objective function has a specific compositional structure.

**Assumption 4.1.** *The objective is $f(x) = h(g(x))$ where:*

- *The function $g : \mathbb{R}^d \to \mathbb{R}$ is twice continuously differentiable, $\nabla g$ is $L_g$-Lipschitz continuous, and $g$ is $\mu_g$-PL with minimizer $x^*$ satisfying $g(x^*) = 0$*

- *The function $h : [0, \infty) \to \mathbb{R}$ is twice continuously differentiable, strictly increasing, strictly convex, $h''$ is non-decreasing, and $h'(t) = \Theta(t^p)$ as $t \to 0^+$, for some $p \geq 1$*

We note that the assumptions on $g$ imply that

$$2\mu_g \, g(x) \leq \|\nabla g(x)\|^2 \leq 2L_g \, g(x), \qquad \forall x \in \mathbb{R}^d, \tag{4.1}$$

where the right-hand inequality follows from (Wright, 2018, eq. (4.1.3)), and that $x^*$ (the minimizer of $g$) is also a minimizer for $f$ with $f(x^*) = h(0)$.

The function $g$ could be, for example $g(x) = \|Ax - b\|_2^2$ for some consistent linear system $Ax = b$, but in general need not be convex. We are most interested in Assumption 4.1 when the outer function $h$ is very flat near 0, such as $h(t) = t^p$ for $p > 2$, although other functions such as $h(t) = t$ are also allowed. In general, this means that $f$ is not PL, as shown by the case $g(x) = x^2$ and $h(t) = t^2$.

Even though $f$ is not PL, we will show that the iterates generated by Algorithm 1 exhibit a *global* linear convergence rate, which is typically only seen for PL functions (for standard GD-type methods). To show this, we will need the following technical results.

**Lemma 4.2.** *Suppose Assumption 4.1 holds and we perform the iteration*

$$x_{k+1} = x_k - \eta_k \nabla g(x_k),$$

*where there exists $\epsilon \in (0, 1/L_g]$ such that $\epsilon \leq \eta_k \leq 2/L_g - \epsilon$ for all $k$. Then*

$$g(x_k) \leq (1 - \mu_g \epsilon(2 - L_g\epsilon))^k g(x_0).$$

*Proof.* This is a generalization of (Karimi et al., 2016, Theorem 1), which proves the case where $\epsilon = 1/L_g$ (i.e. $\eta_k = 1/L_g$ for all $k$). Since $g$ is $L_g$-smooth, we have

$$g(x_{k+1}) \leq g(x_k) - \eta_k \|\nabla g(x_k)\|^2 + \frac{L_g \eta_k^2}{2} \|\nabla g(x_k)\|^2,$$

$$= g(x_k) - \eta_k \left(1 - \frac{L_g \eta_k}{2}\right) \|\nabla g(x_k)\|^2,$$

$$\leq g(x_k) - 2\mu_g \eta_k \left(1 - \frac{L_g \eta_k}{2}\right) g(x_k),$$

where the last inequality holds from (4.1), and so

$$g(x_{k+1}) \leq \left(1 - 2\mu_g \eta_k \left(1 - \frac{L_g \eta_k}{2}\right)\right) g(x_k),$$

or

$$g(x_k) \leq (1 - \mu_g \eta_k (2 - L_g \eta_k))^k g(x_0).$$

The term $\mu_g \eta_k (2 - L_g \eta_k)$ is positive for all $\eta_k \in (0, 2/L_g)$ and maximized for $\eta_k = 1/L_g$, in which case $\mu \eta_k (2 - L_g \eta_k) = \mu_g/L_g \leq 1$. Hence, if $\eta_k \in [\epsilon, 2/L_g - \epsilon]$, then

$$0 < \mu_g \epsilon(2 - L_g\epsilon) \leq \mu_g \eta_k (2 - L_g \eta_k) \leq \mu_g/L_g \leq 1.$$

The result then follows. $\qquad\square$

**Corollary 4.3.** *Suppose the assumptions of Lemma 4.2 hold. Then $\|\nabla f(x_k)\| \to 0$, R-linearly.*

*Proof.* Since $\nabla f(x) = h'(g(x))\nabla g(x)$, we use (4.1) to get

$$\|\nabla f(x_k)\| = h'(g(x_k)) \cdot \|\nabla g(x_k)\| \leq \sqrt{2L_g} \cdot h'(g(x_k)) \cdot \sqrt{g(x_k)},$$

noting that $h'(g(x_k)) > 0$ since $h$ is strictly increasing. Then by Lemma 4.2, we get $g(x_k) \leq g(x_0)$ and so

$$\|\nabla f(x_k)\| \leq \sqrt{2L_g} \cdot h'(g(x_0)) (1 - \mu_g \epsilon(2 - L_g\epsilon))^{k/2} \sqrt{g(x_0)},$$

where we have used that $h'$ is increasing (since $h$ is increasing and convex), and so $\|\nabla f(x_k)\| \to 0$ R-linearly with rate $(1 - \mu_g \epsilon(2 - L_g\epsilon))^{1/2}$. $\qquad\square$

Now for the objective given by Assumption 4.1, we have

$$\nabla f(x) = h'(g(x))\nabla g(x) \quad \text{and} \quad \nabla^2 f(x) = h''(g(x))\nabla g(x)\nabla g(x)^T + h'(g(x))\nabla^2 g(x),$$

and so for the purposes of calculating $L(x, R)$ we estimate

$$\max_{\|s\| \leq R} \|\nabla^2 f(x + s)\| \leq h''(g(x + s))\|\nabla g(x + s)\|^2 + h'(g(x + s))L_g.$$

Using (4.1) and $\|\nabla g(x + s)\| \leq L_g\|s\| + \|\nabla g(x)\|$ we get the LFSO

$$L(x, R) = h'' \left(\frac{[L_g R + \|\nabla g(x)\|]^2}{2\mu_g}\right) [L_g R + \|\nabla g(x)\|]^2 + h' \left(\frac{[L_g R + \|\nabla g(x)\|]^2}{2\mu_g}\right) L_g. \qquad (4.2)$$

Note that $L$ is non-decreasing in $R$ follows since $h'$ and $h''$ are non-decreasing, and $h'' > 0$ (Assumption 4.1).

By observing the form of $L(x, R)$ (4.2), it is natural to consider $R_k = \|\nabla g(x_k)\|$, since these two quantities (i.e. $R$ and $\|\nabla g(x)\|$) always appear together and it is a computable/known value. It is this choice that will give the fast global convergence rate of Algorithm 1.

**Lemma 4.4.** *Suppose Assumption 4.1 holds and we choose $R_k = \|\nabla g(x_k)\|$ and any $\eta > 0$ in Algorithm 1 (with LFSO (4.2)). Then $\widetilde{R}_k = D_k \|\nabla g(x_k)\|$ for some $D_k \in [1, \max(1, \eta/L_g)]$.*

*Proof.* Substituting our choice of $R_k$ in (4.2) we get

$$\widetilde{R}_k = \max \left\{ \|\nabla g(x_k)\|, \frac{\eta h'(g(x_k))\|\nabla g(x_k)\|}{h''\left(\frac{(L_g+1)^2\|\nabla g(x_k)\|^2}{2\mu_g}\right)(L_g+1)^2\|\nabla g(x_k)\|^2 + h'\left(\frac{(L_g+1)^2\|\nabla g(x_k)\|^2}{2\mu_g}\right)L_g} \right\},$$

$$\leq \max \left\{ \|\nabla g(x_k)\|, \frac{\eta h'\left(\frac{\|\nabla g(x_k)\|^2}{2\mu_g}\right)\|\nabla g(x_k)\|}{h'\left(\frac{(L_g+1)^2\|\nabla g(x_k)\|^2}{2\mu_g}\right)L_g} \right\},$$

$$\leq \max \left\{ 1, \frac{\eta}{L_g} \right\} \|\nabla g(x_k)\|,$$

where the last line follows from $h$ strictly convex (so $h'$ is non-decreasing). This gives $D_k \leq \max\{1, \eta/L_g\}$. That $D_k \geq 1$ follows from $\widetilde{R}_k \geq R_k$ (by definition of $\widetilde{R}_k$). $\qquad\square$

We can now show our global linear rate for Algorithm 1 for this specific problem class.

**Theorem 4.5.** *Suppose Assumption 4.1 holds and we choose $R_k = \|\nabla g(x_k)\|$ and $\eta \in (0, 2)$ in Algorithm 1 (with LFSO (4.2)). Then $\|\nabla f(x_k)\| \to 0$ R-linearly.*

*Proof.* From Lemma 4.4, we have

$$L(x_k, \widetilde{R}_k) = h''\left(\frac{(L_g D_k + 1)^2\|\nabla g(x_k)\|^2}{2\mu_g}\right)(L_g D_k + 1)^2\|\nabla g(x_k)\|^2 + h'\left(\frac{(L_g D_k + 1)^2\|\nabla g(x_k)\|^2}{2\mu_g}\right)L_g.$$

Our iteration can be expressed as

$$x_{k+1} = x_k - \frac{\eta\, h'(g(x_k))}{L(x_k, \widetilde{R}_k)}\nabla g(x_k).$$

Since $h$ is convex, we have $h'' \geq 0$ and $h'$ is non-decreasing, so using (4.1) we get

$$L(x_k, \widetilde{R}_k) \geq h'\left(\frac{(L_g D_k + 1)^2\|\nabla g(x_k)\|^2}{2\mu_g}\right)L_g \geq h'\left(\frac{\|\nabla g(x_k)\|^2}{2\mu_g}\right)L_g \geq h'(g(x_k))L_g,$$

which gives

$$\frac{\eta\, h'(g(x_k))}{L(x_k, \widetilde{R}_k)} \leq \frac{\eta}{L_g}. \tag{4.3}$$

Separately, since $h' > 0$ and $h''$ is non-decreasing, we have (for any $t \geq 0$)

$$t \cdot h''(t) \leq \int_t^{2t} h''(s)ds = h'(2t) - h'(t) \leq h'(2t),$$

and so

$$h''\left(\frac{(L_g D_k + 1)^2\|\nabla g(x_k)\|^2}{2\mu_g}\right)(L_g D_k + 1)^2\|\nabla g(x_k)\|^2 \leq 2\mu_g h'\left(\frac{(L_g D_k + 1)^2\|\nabla g(x_k)\|^2}{\mu_g}\right).$$

Since $h'$ is non-decreasing, we then conclude

$$L(x_k, \widetilde{R}_k) \leq (2\mu_g + L_g)h'\left(\frac{(L_g D_k + 1)^2 \|\nabla g(x_k)\|^2}{\mu_g}\right).$$

From (4.1) we then have

$$L(x_k, \widetilde{R}_k) \leq (2\mu_g + L_g)h'\left(\frac{2L_g(L_g D_k + 1)^2 g(x_k)}{\mu_g}\right).$$

Denoting $C_k := 2L_g(L_g D_k + 1)^2/\mu_g$, this means

$$\frac{\eta \, h'(g(x_k))}{L(x_k, \widetilde{R}_k)} \geq \frac{\eta \, h'(g(x_k))}{(2\mu_g + L_g)h'(C_k g(x_k))}.$$

We note that $1 \leq C_k \leq C_{\max} := 2L_g(L_g \max(1, \eta/L_g) + 1)^2/\mu_g$ from $\mu_g \leq L_g$ and Lemma 4.4, respectively. Also, since Algorithm 1 is monotone (Lemma 3.2, noting Assumption 3.1 is implied by Assumption 4.1), we have $f(x_k) \leq f(x_0)$ and so $g(x_k) \leq g(x_0)$ since $h$ is increasing.

Now, since $h'(t) = \Theta(t^p)$ as $t \to 0^+$, there is an interval $[0, \delta]$ constants $0 < C_1 \leq C_2$ for which

$$C_1 t^p \leq h'(t) \leq C_2 t^p, \qquad \forall t \in [0, \delta]$$

If $C_k g(x_k) \leq \delta$, then we have

$$\frac{h'(g(x_k))}{h'(C_k g(x_k))} \geq \frac{C_1 g(x_k)^p}{C_2 C_k^p g(x_k)^p} \geq \frac{C_1}{C_2 C_{\max}^p} > 0.$$

By contrast, if $C_k g(x_k) > \delta$, then $g(x_0) \geq g(x_k) > \delta/C_{\max}$. Since $h'$ is continuous and strictly positive for all $t > 0$, we have

$$\epsilon_0 := \min_{\delta/C_{\max} \leq t \leq g(x_0)} \frac{h'(t)}{h'(C_{\max} t)} > 0,$$

and so in this case we have

$$\frac{h'(g(x_k)}{h'(C_k g(x_k))} \geq \frac{h'(g(x_k))}{h'(C_{\max} g(x_k))} \geq \epsilon_0.$$

In either case, we have

$$\frac{\eta \, h'(g(x_k))}{L(x_k, \widetilde{R}_k)} \geq \frac{\eta}{2\mu_g + L_g} \min\left(\frac{C_1}{C_2 C_{\max}^p}, \epsilon_0\right) > 0. \tag{4.4}$$

The result then follows by combining (4.3) and (4.4) with $\eta \in (0, 2)$ and Corollary 4.3. □

We reiterate that this result is a global linear rate, and does not require $x_0$ to be sufficiently close to $x^*$ (although the actual linear rate does potentially depend on $x_0$). This applies to functions with extremely flat minima, such as $f(x) = \|x\|_2^{2p}$ for any $p \geq 1$ (by taking $g(x) = \|x\|_2^2$ and $h(t) = t^p$).

## 5 Global Rate for Linear Regression

Another important problem class where Algorithm 1 can achieve a global linear rate (at least in some cases), is the case of linear regression with an $\ell_p$ loss function:

$$\min_{x \in \mathbb{R}^d} f(x) := \|Ax - b\|_{2p}^{2p} = \sum_{i=1}^{n} (a_i^T x - b_i)^{2p}, \tag{5.1}$$

for some $A \in \mathbb{R}^{n \times d}$ with rows $a_i \in \mathbb{R}^d$ for $i = 1, \ldots, n$, and $b \in \mathbb{R}^n$, and $p \in \mathbb{N}$. The choice of norm here avoids any issues of non-smoothness in the objective, but taking $p \to \infty$ again recovers a situation with extremely flat local minima. Our theoretical results will hold in the case where $A$ is sufficiently well-conditioned, which in particular includes the case $f(x) = \|x\|_{2p}^{2p}$.

**Assumption 5.1.** *The matrix $A$ has full rank, $n \leq d$, and $\kappa(A)^4 < n/(n-1)$, where $\kappa(A)$ is the 2-norm condition number of $A$.*

Observe that Assumption 5.1 implies the system $Ax = b$ is consistent. We also note that Assumption 5.1 is quite restrictive, especially when $n$ is large, requiring the rows of $A$ to be almost orthonormal. Given this, we delegate the technical details of this section to Appendix B.

For (5.1), we have

$$\nabla f(x) = 2pA^T(Ax - b)^{2p-1} \quad \text{and} \quad \nabla^2 f(x) = 2p(2p-1)A^T \operatorname{diag}(Ax - b)^{2p-2}A,$$

where $(Ax - b)^p$ is understood to represent element-wise powers. We now need to provide an LFSO for $f(x)$ (5.1). In the case $p = 1$—that is, typical linear least-squares regression—we have $\nabla^2 f(x) = 2p(2p-1)A^T A$ and so we automatically get

$$L(x, R) = 2\|A\|_2^2, \tag{5.2}$$

as a valid LFSO. In the case $p = 2, 3, 4, \ldots$ a valid LFSO for (5.1) is

$$L(x, R) = 2p(2p-1)\|A\|_2^2 \cdot 2^{2p-3} \left[ \|Ax - b\|_\infty^{2p-2} + \left( \max_{i=1,\ldots,n} \|a_i\|_2^{2p-2} \right) R^{2p-2} \right], \tag{5.3}$$

as derived in Appendix B.1. We note that substituting $p = 1$ into (5.3) recovers (5.2) and so (5.3) is a valid LFSO for all $p \in \mathbb{N}$. Observing the form of (5.3), a natural choice for $R_k$ is $R_k = \|Ax_k - b\|_\infty$.

**Theorem 5.2.** *Suppose Assumption 5.1 holds, and we choose $R_k = \|Ax_k - b\|_\infty$ and $\eta \in (0, 1]$ in Algorithm 1 (with LFSO (5.3)). Then for any $p \in \mathbb{N}$, the residual $\|r_k\| \to 0$ Q-linearly.*

*Proof.* See Appendix B.2. $\qquad\square$

The above gives a global linear rate under some restrictive assumptions on the problem (5.1). They are satisfied if $A = I$, for example, which gives us the objective $f(x) = \|x\|_{2p}^{2p}$ for any $p$.

# 6 Numerical Experiments

In this section, we provide some brief numerical experiments confirming the global linear rate for Algorithm 1 (with an appropriate choice of $R_k$ and no tuning of the fixed stepsize $\eta = 1$) for objectives of the form $f(x) = \|x\|_2^{2p}$ as in Section 4 and $f(x) = \|x\|_{2p}^{2p}$ as in Section 5. In all cases, we use $d = 10$ dimensional problems and starting point $x_0 = (1, \ldots, 1)^T$ and $p = 1, \ldots, 5$. For both objectives, the $p = 1$ case gives the strongly convex objective $f(x) = \|x\|_2^2$, but for $p > 1$ we only have (non-strong) convexity and a flat neighborhood of the global minimizer $x^* = 0$. In all cases we plot the normalized gradient decrease $\|\nabla f(x_k)\|_2 / \|\nabla f(x_0)\|_2$ as a function of iteration $k$, for up to $10^4$ iterations.

For standard gradient descent with fixed stepsize, we get the results shown in Figure 1. To see sufficiently fast convergence, some mild tuning of the stepsize $\eta$ was required. For both objectives, we see that gradient descent achieves a global linear rate for $p = 1$ as expected, but a clearly sublinear rate for all $p > 1$, again in line with expectations.

When running Algorithm 1, we use $R_k = \|\nabla g(x_k)\|_2 = 2\|x\|_2$ for $f(x) = \|x\|_2^{2p}$ (based on the framework of Section 4) and $R_k = \|x\|_\infty$ for $f(x) = \|x\|_{2p}^{2p}$ (based on the framework of Section 5). The corresponding results are given in Figure 2. Here we see the expected global linear rate for all values of $p$, not just the strongly convex case $p > 1$. We do note however that the rate of convergence is faster for smaller values of $p$.

By comparison, Appendix C.1 shows the same results for the adaptive gradient descent algorithm of (Malitsky & Mishchenko, 2020) (using the recommended starting stepsize $\eta_0 = 10^{-10}$, which is automatically adjusted after the first iteration). This method outperforms both standard gradient descent and Algorithm 1, and indeed appears to show a global linear convergence rate for all problems. However, this result is not proven, and no convergence results for this method are known for nonconvex objectives.

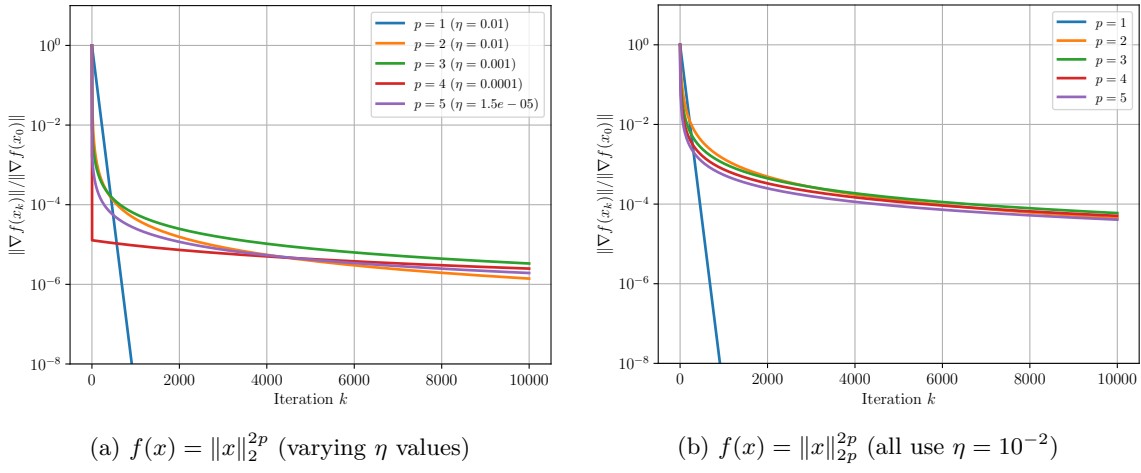

(a) $f(x) = \|x\|_2^{2p}$ (varying $\eta$ values)

(b) $f(x) = \|x\|_{2p}^{2p}$ (all use $\eta = 10^{-2}$)

Figure 1: Global sublinear rate $\|\nabla f(x_k)\| \to 0$ achieved by gradient descent with fixed stepsize for non-strongly convex functions with flat minima. Plots show $\|\nabla f(x_k)\|_2 / \|\nabla f(x_0)\|_2$ as a function of $k$.

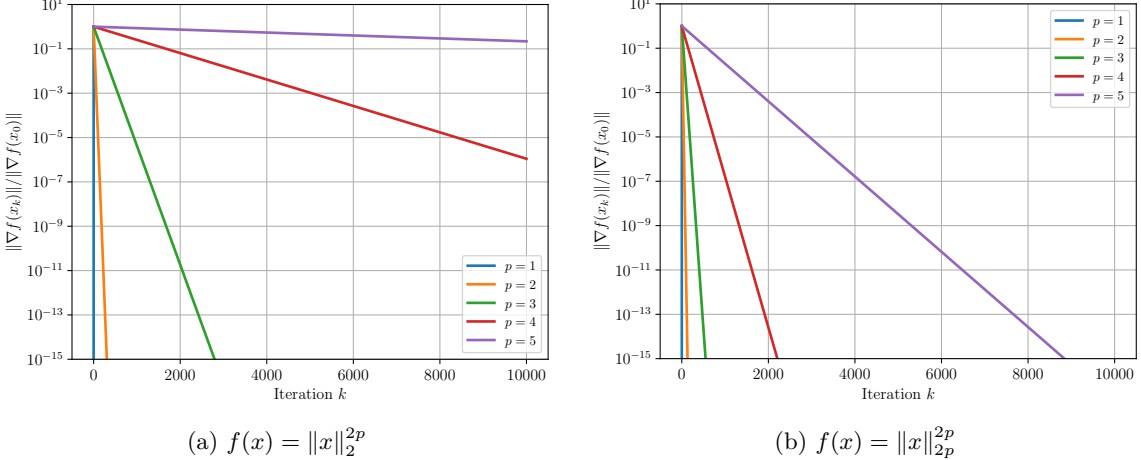

(a) $f(x) = \|x\|_2^{2p}$

(b) $f(x) = \|x\|_{2p}^{2p}$

Figure 2: Global linear rate $\|\nabla f(x_k)\| \to 0$ achieved by Algorithm 1 for non-strongly convex functions with flat minima. Plots show $\|\nabla f(x_k)\|_2 / \|\nabla f(x_0)\|_2$ as a function of $k$.

**Runtime Performance**   Since Algorithm 1 is more computationally expensive than gradient descent on a per-iteration basis (from up to two evaluations of the LFSO per iteration), the above results are shown in terms of runtime (rather than iterations) in Appendix C.2. We see that Algorithm 1 is slower than gradient descent, but not significantly (within a factor of 2 for the slower problems $p = 3, 4, 5$).

**Mis-specification of $R_k$**   The above results for Algorithm 1 assume that $R_k$ is chosen according to the theoretical results in Sections 4 and 5. In Appendix C.3, we consider the impact of significantly mis-specifying $R_k$ (i.e. by changing the order of magnitude, not just adjusting by a constant) in Algorithm 1. We see that using $R_k$ too small appears to have minimal impact on the convergence rate and $R_k$ too large worsens the convergence rate to sublinear.

## 7   Conclusions and Future Work

We have introduced a new oracle for local first-order smoothness, which exists for a wide range of functions and encodes all of the problem information relevant for selecting stepsizes for gradient descent-type methods. Using the LFSO, we introduced a practical gradient descent-type method, and showed global and local

convergence results under reasonable assumptions. We then showed that this method gives global linear rates for some specific (non-strongly) convex functions with degenerate local minima.

There are many potential directions for future study of LFSOs, including automatic differentiation techniques for building LFSOs and understanding how to pick the forcing sequence $R_k$. This would enable more extensive numerical testing of the LFSO approach to ascertain its usefulness on problems of more practical interest. Additionally, the use of LFSOs in broader optimization settings, such as constrained, bilevel, nonsmooth and/or stochastic optimization is an interesting direction for future work.

### Acknowledgments

FR was partially supported by the Australian Research Council through an Industrial Transformation Training Centre for Information Resilience (IC200100022). Albert S. Berahas was partially supported by the Office of Naval Research under award number N00014-21-1-2532.

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

## A  LFSO for Logistic Regression

Here we derive an LFSO suitable for 2-class logistic regression. Following (Murphy, 2012, Chapter 8), the empirical risk minimization problem is

$$\min_{w \in \mathbb{R}^d} f(w) = \sum_{i=1}^{n} [y_i \log(\mu_i(w)) + (1 - y_i) \log(1 - \mu_i(w)))],$$

where $\mu_i(w) = \sigma(w^T x_i)$ for $\sigma(t) = 1/(1 + e^{-t})$, and we have features $x_i \in \mathbb{R}^d$ and labels $y_i \in \{0, 1\}$. The associated gradient and Hessian may be written as

$$\nabla f(w) = X^T(\mu(w) - y), \qquad \text{and} \qquad \nabla^2 f(w) = X^T S(w) X,$$

where $X \in \mathbb{R}^{n \times d}$ and $y \in \mathbb{R}^n$ contain the training data and labels, $\mu(w) \in \mathbb{R}^n$ has entries $\mu_i(w)$, and $S(w) = \text{diag}(\sigma'(w^T x_i)) \in \mathbb{R}^{n \times n}$.

Since $\sigma'(t) \in [0, 1/4]$ for all $t$, we automatically have $\|\nabla^2 f(w)\|_2 \leq \frac{1}{4}\|X\|_2^2$. Thus $\nabla f$ is $L_f$-smooth and so Algorithm 1 with $\eta = 1$ can reduce to standard gradient descent with learning rate $1/L_f$.

If we want to get a tighter LFSO, we start by noting that $\sigma'(t) = \sigma(t)(1 - \sigma(t))$ and so $\sigma''(t) = 2\sigma(t)^3 - 3\sigma(t)^2 + \sigma(t)$, and since $\sigma(t) \in [0, 1]$ we have $|\sigma''(t)| \leq \sqrt{3}/18$ for all $t$. Hence by Cauchy-Schwarz and Taylor's Theorem, we have

$$\sigma'((w + s)^T x_i) \leq \sigma'(w^T x_i) + \frac{\sqrt{3}\, R\|x_i\|_2}{18},$$

whenever $\|s\|_2 \leq R$. Hence a more precise LFSO for this problem is

$$L(w, R) = \|X\|_2^2 \max_{i=1,\ldots,n} \min\left\{ \sigma'(w^T x_i) + \frac{\sqrt{3}\, R \|x_i\|_2}{18}, \frac{1}{4} \right\}. \tag{A.1}$$

If the data $X$ is separable, then near the optimum $w^*$ it is likely that all $w^T x_i$ are far from zero, and so $\sigma'(w^T x_i) \ll 1/4$, and so taking this LFSO with small $R$ may improve on the simpler $L(w, R) = \frac{1}{4}\|X\|_2^2$ from Lipschitz continuous gradients.

The computational cost of evaluating $\nabla f(w)$ is one evaluation of $\mu_i(w) = \sigma(w^T x_i)$ for each $i$ and one matrix-vector product with $X$. If $\|X\|_2^2$ and $\|x_i\|_2$ have been pre-computed, then since $\sigma'(w^T x_i) = \mu_i(w)(1 - \mu_i(w))$, computing the LFSO (A.1) requires the evaluation of each $\mu_i(w)$ plus $\mathcal{O}(n)$ in extra calculations. Thus one LFSO evaluation is actually computationally cheaper than one gradient evaluation.

# B    Technical Details for Section 5

## B.1    Derivation of LFSO

Here we derive the LFSO (5.3) for problem (5.1) with $p = 2, 3, 4, \ldots$.

**Lemma B.1.** *For any $x_1, \ldots, x_m \in \mathbb{R}$ and $t \geq 1$, we have*

$$\left| \sum_{i=1}^{m} x_i \right|^t \leq m^{t-1} \sum_{i=1}^{m} |x_i|^t.$$

*Proof.* If $t = 1$, this is the triangle inequality. For $t > 1$, we use Hölder's inequality to get

$$|x^T e|^t \leq \left( \|x\|_t \|e\|_{t/(t-1)} \right)^t,$$

where $x = (x_1, \ldots, x_m) \in \mathbb{R}^m$ and $e = (1, \ldots, 1) \in \mathbb{R}^m$. The result then follows from $\|e\|_{t/(t-1)}^t = m^{t-1}$.  □

Using Lemma B.1, we get

$$(a_i^T x - b_i + a_i^T s)^{2p-2} \leq 2^{2p-3} \left[ (a_i^T x - b_i)^{2p-2} + (a_i^T s)^{2p-2} \right],$$

for any $p = 2, 3, 4, \ldots$. Noting that $\| \operatorname{diag}(Ax - b)^{2p-2} \|_2 = \|Ax - b\|_\infty^{2p-2}$, for this range of $p$ we get

$$
\begin{aligned}
\max_{y \in B(x,R)} \|\nabla^2 f(y)\|_2 &= \max_{\|s\|_2 \leq R} \|\nabla^2 f(x+s)\|_2, \\
&\leq \max_{\|s\|_2 \leq R} 2p(2p-1)\|A\|_2^2 \max_{i=1,\ldots,n}(a_i^T x - b_i + a_i^T s)^{2p-2}, \\
&\leq \max_{i=1,\ldots,n} \max_{\|s\|_2 \leq R} 2p(2p-1)\|A\|_2^2 \cdot 2^{2p-3}[(a_i^T x - b_i)^{2p-2} + (a_i^T s)^{2p-2}], \\
&\leq \max_{i=1,\ldots,n} 2p(2p-1)\|A\|_2^2 \cdot 2^{2p-3}((a_i^T x - b_i)^{2p-2} + \|a_i\|_2^{2p-2} R^{2p-2}),
\end{aligned}
$$

and so (5.3) is a valid LFSO.

## B.2    Proof of Theorem 5.2

Given the choice $R_k = \|Ax_k - b\|_\infty$, and noting that Algorithm 1 works if $\|\nabla f(x_k)\|$ in (2.3) is replaced by any upper bound for $\|\nabla f(x_k)\|$, we have for $p \geq 2$,

$$
\begin{aligned}
\widetilde{R}_k &= \max\left( \left\{ \|Ax_k - b\|_\infty, \frac{2p\eta\|A\|\sqrt{n}\|Ax_k - b\|_\infty^{2p-1}}{2p(2p-1)\|A\|_2^2 \cdot 2^{2p-3}\left[ \|Ax_k - b\|_\infty^{2p-2} + \left(\max_{i=1,\ldots,n}\|a_i\|_2^{2p-2}\right)\|Ax_k - b\|_\infty^{2p-2} \right]} \right\} \right), \\
&= \max\left\{ \|Ax_k - b\|_\infty, \frac{\eta\sqrt{n}\|Ax_k - b\|_\infty}{(2p-1)\|A\|_2 \cdot 2^{2p-3}\left[ 1 + \left(\max_{i=1,\ldots,n}\|a_i\|_2^{2p-2}\right) \right]} \right\}, \\
&= c_1(A, n, p, \eta)\|Ax_k - b\|_\infty,
\end{aligned}
$$

where

$$c_1(A, n, p, \eta) := \max \left\{ 1, \frac{\eta \sqrt{n}}{(2p-1)\|A\|_2 \cdot 2^{2p-3} \left[ 1 + \left( \max_{i=1,\dots,n} \|a_i\|_2^{2p-2} \right) \right]} \right\}.$$

Thus we have

$$L(x_k, \widetilde{R}_k) = 2p(2p-1)\|A\|_2^2 \cdot 2^{2p-3} \left[ \|Ax_k - b\|_\infty^{2p-2} + \left( \max_{i=1,\dots,n} \|a_i\|_2^{2p-2} \right) c_1(A, n, p, \eta) \|Ax_k - b\|_\infty^{2p-2} \right],$$

$$= 2p \, c_2(A, n, p, \eta) \|Ax_k - b\|_\infty^{2p-2}, \tag{B.1}$$

where

$$c_2(A, n, p, \eta) := (2p-1)\|A\|_2^2 \cdot 2^{2p-3} \left[ 1 + \left( \max_{i=1,\dots,n} \|a_i\|_2^{2p-2} \right) c_1(A, n, p, \eta) \right],$$

again for $p = 2, 3, 4, \dots$. For $p = 1$, since $L(x, R)$ is independent of $R$, we always have $L(x_k, \widetilde{R}_k) = 2p\|A\|_2^2$, or equivalently (B.1) with $c_2(A, n, 1, \eta) := \|A\|_2^2$. Finally, our iteration is

$$x_{k+1} = x_k - \frac{\eta}{c_2(A, n, p, \eta)\|Ax_k - b\|_\infty^{2p-2}} A^T(Ax_k - b)^{2p-1}.$$

Since we assume our linear system is consistent, we know (5.1) has a global minimizer at $f(x^*) = 0$, and so a suitable error metric is the residual, $r_k := Ax_k - b$. Written in terms of the residual, our iteration is

$$r_{k+1} = r_k - \frac{\eta}{c_2(A, n, p, \eta)\|r_k\|_\infty^{2p-2}} AA^T r_k^{2p-1}. \tag{B.2}$$

In the case $p = 1$, the residual iteration (B.2) becomes

$$r_{k+1} = \left[ I - \frac{\eta}{\|A\|_2^2} AA^T \right] r_k,$$

and so $\|r_k\| \to 0$ linearly for all $\eta \in (0, 1]$, as expected.

Instead, if $p \geq 2$, it suffices to consider the case $r_k \neq 0$. Here, (B.2) may be written as

$$r_{k+1} = \left[ I - \frac{\eta}{c_2} AA^T \right] r_k + \frac{\eta}{c_2} AA^T \left[ r_k - \frac{r_k^{2p-1}}{\|r_k\|_\infty^{2p-2}} \right], \tag{B.3}$$

dropping the arguments $c_2 = c_2(A, n, p, \eta)$ for brevity. To handle the nonlinearity in the second term, we first look at the difference

$$e_k := r_k - \frac{r_k^{2p-1}}{\|r_k\|_\infty^{2p-2}}.$$

Looking at $e_k$ in terms of each component separately, and writing $|[r_k]_i| = \|r_k\|_\infty / \alpha_{k,i}$ for some $\alpha_{k,i} \geq 1$ (with $\alpha_{k,i} = \infty$ if $[r_k]_i = 0$), we get

$$[e_k]_i = \left( 1 - \frac{1}{\alpha_{k,i}^{2p-2}} \right) [r_k]_i.$$

Note specifically that $\alpha_{k,i} = 1$ for the index $i$ for which $|[r_k]_i| = \|r_k\|_\infty$. Then for any $M > 1$, we have

$$\|e_k\|_2^2 = \sum_{i:\alpha_{k,i} \geq M} \left( 1 - \frac{1}{\alpha_{k,i}^{2p-2}} \right)^2 [r_k]_i^2 + \sum_{i:\alpha_{k,i} < M} \left( 1 - \frac{1}{\alpha_{k,i}^{2p-2}} \right)^2 [r_k]_i^2,$$

$$\leq \sum_{i:\alpha_{k,i} \geq M} [r_k]_i^2 + \sum_{\alpha_{k,i} < M} \left( 1 - \frac{1}{M^{2p-2}} \right)^2 [r_k]_i^2,$$

$$= \|r_k\|_2^2 - \sum_{i:\alpha_{k,i} < M} \left( \frac{2}{M^{2p-2}} - \frac{1}{M^{4p-4}} \right) [r_k]_i^2.$$

Since $M > 1$, there is at least one $i$ with $\alpha_{k,i} < M$ (the index corresponding to $\|r_k\|_\infty$). So,

$$\sum_{i:\alpha_{k,i}<M} [r_k]_i^2 \geq \|r_k\|_\infty^2 \geq \frac{1}{n}\|r_k\|_2^2,$$

using standard norm equivalences. All together, we have

$$\|\widetilde{r}_{k+1}\|_2^2 \leq \left[1 - \frac{1}{n}\left(\frac{2}{M^{2p-2}} - \frac{1}{M^{4p-4}}\right)\right]\|r_k\|_2^2,$$

for any $M > 1$. This bound is tightest as $M \to 1^+$, with $\frac{2}{M^{2p-2}} - \frac{1}{M^{4p-4}} \to 1^-$ in this case. So by taking $M$ sufficiently close to 1, we get

$$\|\widetilde{r}_{k+1}\|_2 \leq (1 - \epsilon)^{1/2}\|r_k\|_2,$$

for any $\epsilon < 1/n$. Returning to (B.3), we get

$$\|r_{k+1}\|_2 \leq \left[\left\|I - \frac{\eta}{c_2}AA^T\right\|_2 + \frac{\eta\|A\|_2^2}{c_2}(1 - \epsilon)^{1/2}\right]\|r_k\|_2,$$

$$\leq \left[\|I - \eta B\|_2 + \eta\|B\|_2(1 - \epsilon)^{1/2}\right]\|r_k\|_2,$$

where $B := AA^T/c_2$ has $\|B\|_2 \leq 1$ since $c_2 \geq \|A\|_2^2$. Thus $\|r_k\|_2 \to 0$ Q-linearly provided $\|I - \eta B\|_2 + \eta\|B\|_2(1 - \epsilon)^{1/2} < 1$.

Defining $\sigma_i > 0$ as the $i$th singular value of $B \in \mathbb{R}^{n \times n}$, since $\|B\|_2 \leq 1$ and $B$ is full rank (since $A$ is full rank and $n \leq d$) we know $0 < \sigma_n \leq \sigma_1 \leq 1$. Since $\eta \in (0, 1]$ by assumption, we have

$$\|I - \eta B\|_2 + \eta\|B\|_2(1 - \epsilon)^{1/2} = \max\{|1 - \eta\sigma_1|, |1 - \eta\sigma_n|\} + \eta(1 - \epsilon)^{1/2}\sigma_1,$$

$$= (1 - \eta\sigma_n) + \eta(1 - \epsilon)^{1/2}\sigma_1.$$

This factor is in $(0, 1)$ provided

$$\eta\sigma_n - \eta(1 - \epsilon)^{1/2}\sigma_1 \in (0, 1).$$

Since $\eta, \sigma_n \leq 1$, this holds if $\sigma_n > (1 - \epsilon)^{1/2}\sigma_1$. Since $\kappa(A)^2 = \kappa(B) = \sigma_1/\sigma_m$, this is equivalent to $\kappa(A)^4 < 1/(1 - \epsilon)$. Since $\epsilon < 1/n$ is arbitrary, this holds from $\kappa(A)^4 < n/(n - 1)$ (Assumption 5.1).

## C   Additional Numerical Results

### C.1   Adaptive Gradient Descent

The below shows the convergence of adaptive gradient descent (Malitsky & Mishchenko, 2020) on the same test problems as Section 6.

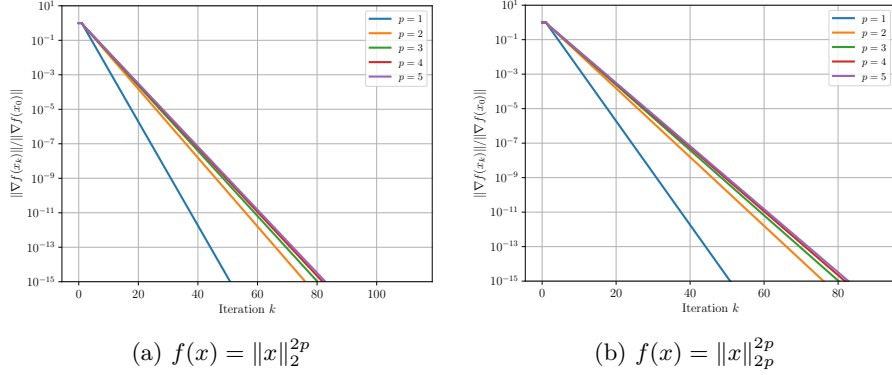

(a) $f(x) = \|x\|_2^{2p}$        (b) $f(x) = \|x\|_{2p}^{2p}$

Figure 3: Convergence $\|\nabla f(x_k)\| \to 0$ achieved by adaptive gradient descent (Malitsky & Mishchenko, 2020) for non-strongly convex functions with flat minima. Plots show $\|\nabla f(x_k)\|_2/\|\nabla f(x_0)\|_2$ as a function of $k$.

## C.2 Runtime Performance

In the below we compare the gradient decrease achieved by gradient descent (fixed $\eta$) and Algorithm 1 as in Section 6, but measured by runtime (total CPU time) rather than iterations.

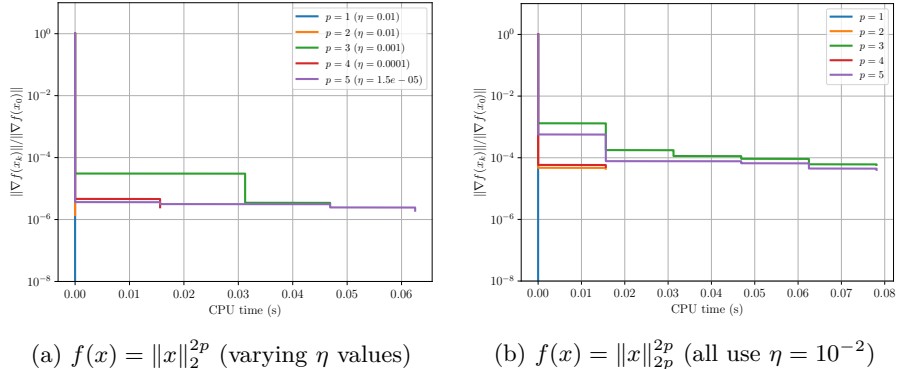

(a) $f(x) = \|x\|_2^{2p}$ (varying $\eta$ values)    (b) $f(x) = \|x\|_{2p}^{2p}$ (all use $\eta = 10^{-2}$)

Figure 4: Convergence rate $\|\nabla f(x_k)\| \to 0$ achieved by gradient descent with fixed stepsize for non-strongly convex functions with flat minima. Plots show $\|\nabla f(x_k)\|_2/\|\nabla f(x_0)\|_2$ as a function of CPU time.

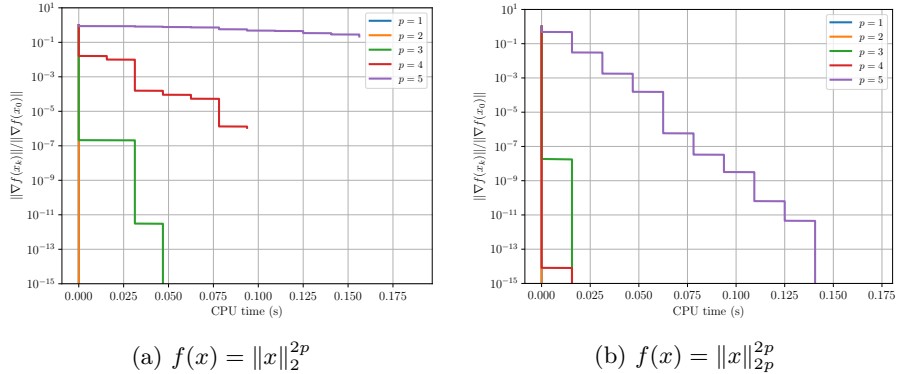

(a) $f(x) = \|x\|_2^{2p}$    (b) $f(x) = \|x\|_{2p}^{2p}$

Figure 5: Convergence rate $\|\nabla f(x_k)\| \to 0$ achieved by Algorithm 1 for non-strongly convex functions with flat minima. Plots show $\|\nabla f(x_k)\|_2/\|\nabla f(x_0)\|_2$ as a function of CPU time.

## C.3 Sub-optimal choice of $R_k$

The results for Algorithm 1 presented in Section 6 use the choices of $R_k$ given in Sections 4 and 5 (namely $R_k^{\mathrm{optimal}} = \|\nabla g(x_k)\|_2 = 2\|x\|_2$ for $f(x) = \|x\|_2^{2p}$ and $R_k^{\mathrm{optimal}} = \|x\|_\infty$ for $f(x) = \|x\|_{2p}^{2p}$). Here we consider the performance of Algorithm 1 when $R_k$ is significantly mis-specified (i.e. change in the order of $R_k$, not just constants).

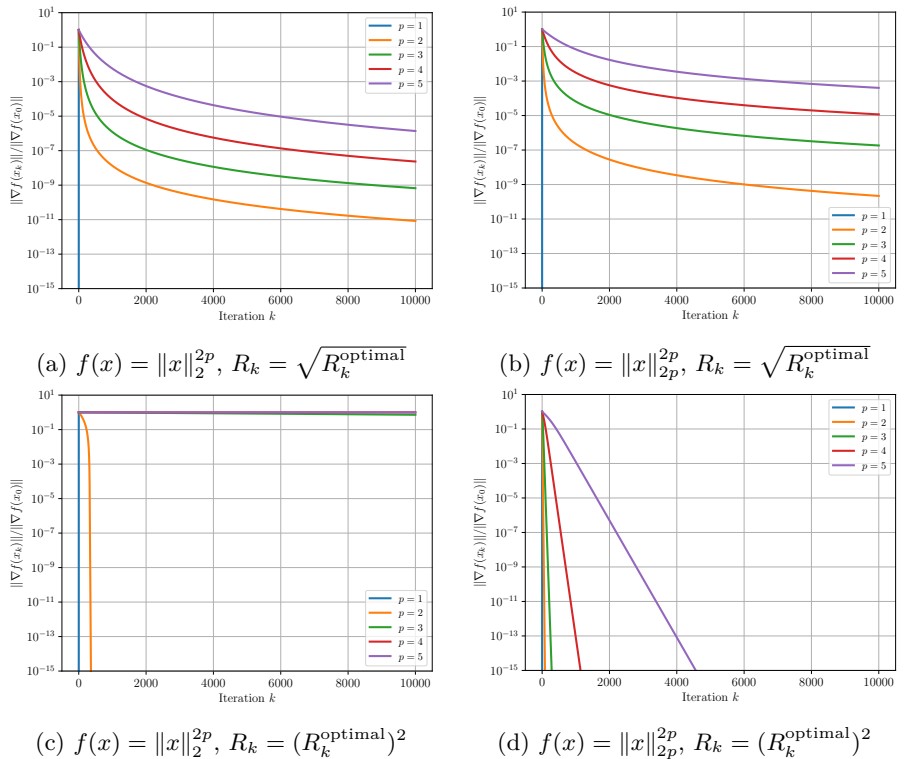

Figure 6: Performance of Algorithm 1 with sub-optimal choices of $R_k$. Plots show $\|\nabla f(x_k)\|_2/\|\nabla f(x_0)\|_2$ as a function of $k$.

In Figure 6c for $p \in \{3, 4, 5\}$, we note that the significant reduction in convergence rate is likely because the new $R_k$ is initially significantly too large. For $p \in \{1, 2\}$ we are in the regime where $R_k$ is too small (as $x \to 0$), and this appears to have minimal impact on the speed of convergence.

