# OpenReview forum: "Non-Uniform Smoothness for Gradient Descent"
_TMLR — Accepted by TMLR_

### Review · Reviewer_GZSs · 2023-12-11

**Summary Of Contributions:**

The submission presents Local First-Order Smoothness Oracle (LFSO) for gradient descent-type methods. The LFSO allows faster convergence to arbitrarily flat minima of convex functions. The paper demonstrates the effectiveness of the LFSO approach through theoretical analysis and numerical experiments.

**Audience:**

Yes

**Claims And Evidence:**

No

**Requested Changes:**

see the weakness part above

**Strengths And Weaknesses:**

Strengths:

- LFSO allows for the construction of a convergent gradient descent-type iteration without requiring much hyperparameter tuning.

- The LFSO method demonstrates promise in achieving a global linear convergence rate to arbitrarily flat minima of convex functions

- It also presents global convergence results for both nonconvex and PL/strongly convex cases


Weaknesses:

- It needs discussion or analysis on the potential limitations and practical considerations of implementing the LFSO method in real-world applications. The current numerical experiments are not so convincing since only toy examples are used while not considering real experiments

- It could benefit from a more extensive comparison with existing methods, particularly in addressing the trade-offs and potential challenges in using the LFSO approach. It lacks such comparison with other competitors, which is one of its major drawbacks of this submissions

- It would also be valuable to provide insights into the computational complexity and scalability of the proposed method, especially in comparison to other state-of-the-art algorithms

---

> ### Author Response · Authors · 2024-01-01
>
> Thank you for the feedback. We have uploaded a new version of the manuscript.
>
> For point 1, we agree and have adapted the introduction to emphasize that this work is to introduce and motivate the LFSO approach as an interesting direction for hyperparameter-free optimization, and there are several aspects of future work that are important for making this approach more practical.
>
> We have updated the introduction to give more detailed comparisons with existing hyperparameter-free approaches, and included Appendix B.1 which gives numerical results for the most relevant of these approaches.
>
> We have added more information to the introduction about the complexity/scalability of our approach, noting that it is heavily dependent on the availability of practical LFSOs, an important direction for future work.

---

### Review · Reviewer_F8Co · 2023-12-11

**Summary Of Contributions:**

Consider the problem of unconstrained convex differentiable optimization. This paper proposes a "local first-order smoothness oracle," a "trust-region" generalization of model proposed in Mai et al. (2021). The paper then proposes a gradient descent method and analyzes its global convergence and local convergence rate. As a result, for instance, a linear convergence rate is achieved for minimizing $f (x) = \Vert x \Vert_2^{2p}$ and $g (x) = \Vert x \Vert_{2p}^{2 p}$ when $p$ is large.

**Audience:**

Yes

**Broader Impact Concerns:**

N/A. This is an optimization theory work.

**Claims And Evidence:**

Yes

**Requested Changes:**

Please address the weaknesses above and make necessary modifications to the paper.

**Strengths And Weaknesses:**

## Strengths

1. The smoothness condition and algorithm are new.
2. The analyses are simple yet effective.

## Weaknesses

1. In practice, evaluating $L (x, R)$ can be challenging. This limits the applications of the algorithm. While the paper provides some examples such as the problems of minimizing $f$ and $g$ defined above, these problems are not very relevant in practice. More realistic examples would be beneficial.
2. The proposed algorithm does not specify how to choose the radii $R_k$, yet the convergence and rate guarantees assume specific conditions on $R_k$ and $L ( x_k, R_k )$. Please either demonstrate the difficulty of efficient optimization when these conditions are not met, or provide a rule for selecting $R_k$ (possibly with modifications to the algorithm) that ensures these conditions are met.
5. Assumption 5.1, which requires the condition number of the matrix $A$ to be very close to $1$, is highly restrictive. While the authors explicitly acknowledge this limitation, it does not fully justify the significance of the example.
6. The problem setup, unconstrained differential convex optimization, is fundamental. Please consider generalizations to the constrained optimization setup, composiptimization setup, and possibly mirror descent-type methods.
7. Minor comments:
    - Emphasizing that the problem of minimizing $f$ provides the worst-case bounds for accelerated gradient descent can be misleading, because this paper considers a different oracle model.
    - Coercivity is sufficient to ensure boundedness of the level sets.
    - Theorem 3.7 needs a clearer statement. Please define explicitly how close $x_0$ should be to $x^\star$ and provide the explicit convergence rate (as in the last equation of the proof).

---

> ### Author Response · Authors · 2024-01-01
>
> Thank you for the feedback. We have uploaded a new version of the manuscript.
>
> For point 1, our goal was to introduce a novel smoothness condition/oracle and associated algorithm, motivating this as an interesting direction for hyperparameter-free optimization. Indeed, calculating LFSOs for more complex problems (with automatic differentiation techniques) is an important future direction, and would enable the use of our approach in more realistic settings.
>
> For point 2, you are correct that the choice of initial radii $R_k$ is another important future direction. For convergence guarantees, Corollary 3.4 suggests that the requirements on $R_k$ are quite weak (boundedness in that case) and the condition on $L(x_k,R_k)$ is achieved. To further study this, in Appendix B.3 we repeat our numerical experiments with the $R_k$ mis-specified (in the sense that they are not chosen according to the theory in Sections 4 and 5). This does not affect whether or not convergence occurs, but does impact the rate of convergence (ultimately suggesting that underestimating the optimal $R_k$ is better than overestimating).
>
> For point 3, we agree that this is a restrictive assumption; our goal was to show another example where global linear convergence to flat minima is achieved. To decrease the emphasis of Section 5, we have moved most of the technical details to Appendix A, significantly shortening the main text.
>
> For point 4, we agree that these are all useful directions to generalize our approach, but we think this is better as future work. Our goal here is motivate the LFSO approach by introducing key concepts/results and showing some interesting initial results (with this already being a `long submission' by TMLR standards). There are important challenges around calculating LFSOs and selecting $R_k$ that are important first steps for generalizing our approach.
>
> For point 5, we agree and have updated the text in Section 3 (2nd and 3rd points) and the abstract/introduction for the first point (reducing the emphasis on out-performing true first-order methods).

---

> > ### Comment · Reviewer_F8Co · 2024-01-11
> >
> > Thank you for your response. I believe this work explores an interesting direction but appears somewhat premature. I encourage the authors to identify and work out relevant applications to demonstrate the significance of the proposed framework. Regarding the acceptance criteria for TMLR:
> > - The claims in the abstract are now supported by the results in the paper, as the authors have removed the last sentence from the abstract.
> > - I am uncertain whether TMLR's audience would find this work engaging, given that the authors only provided toy examples with restrictive assumptions.
> >
> > Consequently, my evaluation leans towards a borderline accept.

---

### Review · Reviewer_Rqa4 · 2023-12-21

**Summary Of Contributions:**

The authors propose an adaptive and parameter-free algorithm based on the on-the-fly estimation of the local Lipschitz constant. They show linear convergence of the proposed algorithm under the Polyak-Lojasiewicz (PL, Thm 3.6) and local strong convexity (Thm 3.7). These results are then extended to compositional functions $f(x) = h(g(x))$ (Thm 4.5) and extended to the specific case of $l_p$ linear regressions with lighter hypotheses (Thm 5.3).

**Audience:**

Yes

**Claims And Evidence:**

No

**Requested Changes:**

- Clear cost per step of the proposed algorithm
- Clear explanation of how $\eta$ is tuned
- Comparison with other adaptive algorithms: theoretically on the rates and empirically (again I am not asking for SOTA algorithm, just a fair scientific comparison for insights, it is OK if the proposed algorithm is slower)
- Wall clock time for the experiments, or fair computational comparison
- Make experiments and theory match: either remove Parts 3 and 4, or add experiments in these settings

**Strengths And Weaknesses:**

Strengths:
- Adaptive and parameter-free algorithms are critical to obtain easy-to-use algorithms for practitioners.

Weaknesses:
- It seems that Parts 3 and 4 are not correlated with the experiments: experiments only show linear convergence of the proposed algorithm on $l_p$ linear regression. To be more explicit, I do not see the theoretical or practical advantage of the proposed algorithm for functions satisfying the PL hypothesis.

Questions:
- Can you ensure that the theoretical rate obtained in Thm 3.6 under the PL condition is better than the one of gradient descent? Or that the step size is larger than $1/L$ at each step?

- What's the point of Thm 3.7?
The two main convergence results are under PL (Thm 3.6) or strong local convexity (Thm 3.7). I was wondering if the authors knew the relationship between the two hypotheses. Is PL weaker or stronger than strong convexity?
If the goal of Thm3.7 is to obtain linear convergence on the iterates, it seems that the PL condition is sufficient; see, for instance, Thm. 4.1 of [1]

- Does Assumption 5.1 imply local strong convexity? PL?

- What is the exact computational complexity of the proposed algorithm? The only sentence I found is rather vague: "it requires one evaluation of $\nabla f$ and possibly two evaluations of $L$".
What is the cost of evaluating $L$? Could you add these computational costs more transparently directly in Algorithm 1? There is no mention of how to tune the hyperparameter $\eta$ in the experiments: do you have to tune it in practice?
How costly is the computation of $L$ in practice? Could you add the same graph with the wall clock time?

- I am not asking for exhaustive SOTA experiments, but adding other adaptive algorithms, such as those mentioned in the introduction, would significantly strengthen the paper.

- A setting where adaptivity is very much needed is bilevel optimization, which has significant similarities with compositional optimization. Could you imagine extending your results to this setting?

[1] Convergence of the forward-backward algorithm: beyond the worst-case with the help of geometry

---

> ### Author Response · Authors · 2024-01-01
>
> Thank you for the feedback. We have uploaded a new version of the manuscript.
>
> Regarding the weakness mentioned, Section 4 is used for the numerical experiments. The goal of Section 3 is to provide standard convergence results similar to what one might prove for gradient descent, essentially showing how LFSO generalizes gradient descent with $1/L$ stepsizes. The results in Sections 4 and 5 give some examples where the LFSO approach gives interesting results. Although these are for special cases, we think our results as a whole motivate further analysis of LFSO-based methods, particularly determining where LFSOs may be easily obtained and good choices of sequences $R_k$ (see future work in Section 7).
>
> Our PL rate in Theorem 3.6 is not guaranteed to improve on gradient descent, since $L(x,R)=L_f$ for $L_f$-smooth functions is a valid LFSO, and all our results then recover standard GD theory. Our approach is more general, not being restricted to the $L_f$-smooth case.
>
> The two properties PL and $\lambda_{\min}(\nabla^2 f(x^*))>0$ are  independent of each other. For example, $f(x_1,x_2)=x_1^2$ is PL since it is of the form $g(Ax)$ for strongly convex $g$ but only has degenerate minima. On the other hand, many local minima of nonconvex functions are non-degenerate, such as $\sin(x)$, but PL implies all stationary points are global minimizers. Theorem 3.7 is a useful local convergence result for general nonconvex optimization. Similarly, Assumption 5.1 is independent of these two conditions: $f(x)=\|x\|_{2p}^{2p}$ is not PL and the minimizer is degenerate for $p>1$.
>
> We have added more information about the cost of the proposed algorithm to the introduction, but ultimately it depends on the cost of evaluating an LFSO (a key aspect to the future work). For our test examples, we have added runtime tests (Appendix B.2), which shows the LFSO approach is approximately 2 times slower than GD.
>
> The advantage of our approach is that there is no hyperparameter tuning required, once an LFSO (and $R_k$) is chosen. All our numerical results use $\eta=1$ and the initial $R_k$ choice (line 3 of Algorithm 1) based on the theory in Sections 4 and 5. We have clarified this in Section 6.
>
> We have added a more detailed comparison with other methods in the introduction, in particular noting that our approach is the most general, since it applies to any (nonconvex) $C^2$ function with an LFSO, where most approaches require convexity (and in some cases the objective itself being Lipschitz continuous). For our numerical experiments, the most relevant approach is that of Malitsky & Mishchenko (2020), and we have added numerical results in Appendix B.1. Their approach outperforms LFSO and also appears to have global linear convergence, but this is not proven, and their method only converges for convex functions.
>
> Thank you for the suggestion to consider bilevel optimization. Certainly it would be interesting to consider LFSOs for lower-level problems (i.e. parametric optimization), where potentially the parameter is another input to the LFSO. We have included this as another potential future direction in Section 7.

---

### Review · Reviewer_wNUz · 2023-12-23

**Summary Of Contributions:**

This paper considers the problem of non-convex optimization with a local smoothness oracle. The paper relaxes the local smoothness assumption of [1] to only hold over a ball of a radius $R$ rather than pointwise. This introduces the problem of choosing which radius value $R$ to query the local smoothness oracle at. The paper gives an algorithm (Algorithm 1, p. 4) in which they set the radius $R$ by following a simple rule (eqn. 2.3).  The authors prove global convergence (with the ordinary $\epsilon^{-2}$ rate) in Corollary 3.4, and give a similar result for functions satisfying the PL condition in Theorem 3.6. They show the algorithm converges linearly if initialized within a small neighborhood of the minimum in Theorem. 3.7. They also give a result for a composition of a PL function and a strictly convex function with monotone second derivatives (see Assumption 4.1) in Theorem 4.5. Finally, an important application of this algorithm is given in Section 5, where it is applied to linear regression of the form $f(x) = \sum_{i=1}^n (a_i^{\top} x - b_i)^{2p}$. Under the assumption that the data matrix is well conditioned, Theorem 5.3 gives linear convergence for Algorithm 1 on this function class for any value of $p$.

[1] Jincheng Mei, Yue Gao, Bo Dai, Csaba Szepesvari, and Dale Schuurmans. Leveraging non-uniformity in first-order non-convex optimization. In Proceedings of the 38th International Conference on Machine Learning,2021.

**Audience:**

Yes

**Broader Impact Concerns:**

Not applicable.

**Claims And Evidence:**

Yes

**Requested Changes:**

Please address my concerns above, particularly in point (4).

**Strengths And Weaknesses:**

1. (Strength) The algorithm given by the paper is quite simple, and its analysis is straightforward. The new condition is a clear generalization of prior work.
2. (Strength) The obtained rate only depend on the local smoothness constants rather than the global constant, therefore they can be significantly tighter than the standard rate for GD. Although how much tighter they are is not clear.
3. (Weakness) The points queried by algorithm 1 always happen to be queried at nondecreasing radiuses $R_k$-- this means that, for example, if the smoothness near the optimum is very small, we may still end up getting a large value for it from the oracle because our query radius $R_k$ is very large and has never decreased.
4. (Weakness) It is well-known that regression problems of the form $\sum_{i=1}^n (a_i^{\top} x - b_i)^p$ can be solved by Iteratively reweighted least squares (IRLS) for any $p$. IRLS has the same per-iteration cost as gradient descent. Moreover, it does not require any access to a local smoothness oracle or global parameters. There should both a comparison in theory and in experiments between the proposed algorithm and IRLS for solving this problem.
5. (Minor point) While it is true that the algorithm is technically hyperparameter-free, this information is baked into the (strong) assumption of access to such an oracle. Are there any examples where this oracle is available, other than for standard linear regression?

---

> ### Author Response · Authors · 2024-01-01
>
> Thank you for the feedback. We have uploaded a new version of the manuscript.
>
> For point 3, Algorithm 1 does not require that the $R_k$ values be nondecreasing. The adjustment in line 3 (which ultimately means $\tilde{R}_k\geq R_k$ is used) is required for Lemma 3.2, but we are allowed to decrease $R_k$ across iterations.
>
> For example, in Section 3 we use $R_k=\|\nabla g(x_k)\|$ and so $\lim_{k\to\infty} R_k=0$ (and in this case, the adjustment gives $\tilde{R}_k \leq \max(1,\eta/L_g) R_k \to 0$ by Lemma 4.4). This is what enables the global linear convergence rate, that we consider the objective function's smoothness only very close to the current iterate.
>
> For point 4 on IRLS, we agree that in practice IRLS is a useful method for $\ell_p$ regression problems. It typically requires solving one linear least-squares problem per iteration, which is more expensive than gradient descent/LFSO (which requires the equivalent of two matrix-vector products per iteration), and indeed for our simple numerical experiments converges in exactly 1 iteration. We have added a discussion about IRLS including relevant convergence theory to the introduction.
>
> For point 5, we agree that our approach relies on the availability of an LFSO. Our goal here was to illustrate that LFSO has potential for hyperparameter-free optimization. The systematic derivation of LFSOs through automatic differentiation techniques is an important avenue of future work to enable this approach (mentioned in Section 7).

---

> > ### Comment · Reviewer_wNUz · 2024-01-07
> >
> > Thank you for your response. I am still not really convinced of the potential for LFSO in hyperparameter-free optimization, because you're just encapsulating the parameter information in a stronger oracle that you won't have access to in practice. The discussion on the stochastic setting also leaves much to be desired, since while there is one method that requires taking differences of stochastic gradients, most do not. In any case, I appreciate the discussion on IRLS and will recommend the paper for acceptance.

---

### Decision · Action_Editor_rtKx · 2024-01-22

**Recommendation:** Accept with minor revision

**Comment:**

See above

**Audience:**

To help transparent decision-making, I include here the TMLR's criteria:

- Are the claims made in the submission supported by accurate, convincing, and clear evidence?
- *Would at least some individuals in TMLR's audience be interested in knowing the findings of this paper?*

As stated in the "Claims and Evidence" section, while one reviewer found the topic super interesting, another reviewer raised a concern whether TMLR's audience would find this work engaging, given that the authors only provided toy examples with restrictive assumptions.

**Claims And Evidence:**

To help transparent decision-making, I include here the TMLR's criteria:

- *Are the claims made in the submission supported by accurate, convincing, and clear evidence?*
- Would at least some individuals in TMLR's audience be interested in knowing the findings of this paper?

A reviewer mentions that the topic is super interesting. Yet, the computational complexity of the proposed method is still unclear to the reviewer, concluding that maybe the authors could add details on simple examples such as logistic regression, or the proposed $\ell_p$-norm minimization.

Finally, one reviewer did not find the current submission has sufficient comparisons with other state-of-the-arts and the results can be improved.

Based on the above discussions, it is clear that the paper passes the bar of acceptance, but might require minor revisions to fully meet the criteria of acceptance at TMLR, which leads to the decision for this submission.

---

> ### Author Response · Authors · 2024-02-11
>
> Thank you for this feedback and the acceptance. In the final version of the manuscript, we have added a detailed discussion about the computational cost of the new method (Section 2 and new Appendix A), and show that the extra oracle evaluations have comparable/cheaper cost than a gradient evaluation (while eliminating the need for expensive hyperparameter tuning). Our numerical testing includes comparisons with the most relevant state-of-the-art method, and future work to evaluate the new oracle using automatic differentiation techniques would enable thorough comparisons on problems of more practical interest. We have clarified this point in the introduction and conclusion.